# Winter Water Quality Modeling in Xiong’an New Area Supported by Hyperspectral Observation

**DOI:** 10.3390/s23084089

**Published:** 2023-04-18

**Authors:** Yuechao Yang, Donghui Zhang, Xusheng Li, Daming Wang, Chunhua Yang, Jianhua Wang

**Affiliations:** 1National Key Laboratory of Remote Sensing Information and Imagery Analyzing Technology, Beijing Research Institute of Uranium Geology, Beijing 100029, China; yangyuechao@briug.cn (Y.Y.); saintlxs@foxmail.com (X.L.); 2Aerospace Information Research Institute, Chinese Academy of Sciences, Beijing 100094, China; wangjh@aircas.ac.cn; 3Tianjin Centre of Geological Survey, China Geological Survey, Tianjin 300170, China; wangdaming@mail.cgs.gov.cn; 4Chongqing Academy of Ecology and Environmental Science, Chongqing 401147, China; yang-chh@163.com

**Keywords:** hyperspectral imager, UAV remote sensing, water quality modeling, Xiong’an New Area, hyperspectral remote sensing, GaiaSky-mini2-VN

## Abstract

Xiong’an New Area is defined as the future city of China, and the regulation of water resources is an important part of the scientific development of the city. Baiyang Lake, the main supplying water for the city, is selected as the study area, and the water quality extraction of four typical river sections is taken as the research objective. The GaiaSky-mini2-VN hyperspectral imaging system was executed on the UAV to obtain the river hyperspectral data for four winter periods. Synchronously, water samples of COD, PI, AN, TP, and TN were collected on the ground, and the in situ data under the same coordinate were obtained. A total of 2 algorithms of band difference and band ratio are established, and the relatively optimal model is obtained based on 18 spectral transformations. The conclusion of the strength of water quality parameters’ content along the four regions is obtained. This study revealed four types of river self-purification, namely, uniform type, enhanced type, jitter type, and weakened type, which provided the scientific basis for water source traceability evaluation, water pollution source area analysis, and water environment comprehensive treatment.

## 1. Introduction

China set up Xiong’an New Area near the capital Beijing in April 2018, known as the country’s millennium plan and national event [1,2]. This new city will play an exemplary role in China’s modernization construction in the future [3]. Therefore, the planning and construction of the Xiong’an New Area are of great significance and far-reaching influence [2,4]. The starting area of the city is about 100 km^2^, the medium-term development area is about 200 km^2^, and the long-term control area is about 2000 km^2^ [3]. Baiyang Lake, located in Xiong’an New Area, is a national key tourist and open area, with a water area of 366 km^2^, and it is the largest freshwater lake in the North China Plain [1]. This lake is formed by the repeated evolution from sea to lake and from lake to land. Its water quality is one of the main ecological and environmental factors for the construction of the new area, and it is urgent to introduce advanced technology and methods. As an important technical means of water environment monitoring, imaging spectrum technology has made great progress in monitoring effects with the continuous promotion of economic and social development since the invention of photography in the 19th century [5].

As an emerging urban construction project, Xiong’an New Area faces a series of challenges in its water quality monitoring [6]. On the one hand, located at the junction of Beijing, Tianjin, and Hebei, the surrounding industrial and agricultural activities have a relatively large impact on the water environment, requiring a considerable investment of manpower and resources for comprehensive protection and monitoring of water resources. On the other hand, due to the relatively limited fiscal expenditures in the new area, how to allocate and utilize monitoring and protection resources reasonably is also a challenge [7]. Traditional water quality monitoring methods require on-site sampling and laboratory testing, which is tedious and time consuming [8,9]. Additionally, given the large scale and numerous water bodies in the new area, how to conduct rapid and efficient monitoring is another challenge. The water quality monitoring data is relatively large, and comprehensively analyzing water quality information through data modeling, data fusion, and other technical means and providing scientifically effective strategies and suggestions is another technical challenge [10,11]. As Xiong’an New Area is rapidly developing and expanding, the extent to which residents and businesses attach importance to environmental protection varies, and how to strengthen publicity and education and increase public participation to guide everyone to actively participate in environmental protection is an important challenge.

Firstly, traditional methods cannot meet the demand for efficient water quality monitoring. Currently, traditional water quality monitoring methods in Xiong’an New Area mainly include on-site sampling, sensor monitoring, GIS-based water quality evaluation, and remote-sensing monitoring [12]. On-site sampling requires collecting water samples and analyzing them in a laboratory, which is time consuming, requires a lot of manpower, and has limited coverage [13]. Sensor monitoring involves installing a large number of sensors in water bodies to monitor parameters such as dissolved oxygen, pH value, and temperature in real time, but this method has high management and maintenance costs and limited coverage [14,15,16]. GIS technology is used to divide water bodies in Xiong’an New Area into zones, select appropriate water quality indicators based on their characteristics, establish a data model, and conduct quantitative analysis to evaluate the water quality of each zone [17,18]. Traditional remote-sensing technology can obtain surface spectral data of water bodies and infer the chemical composition of water bodies by calculating the absorption and scattering characteristics of chemicals. Although it can effectively improve the accuracy of water quality monitoring by integrating spectral information from countless bands, it still faces challenges such as high monitoring difficulty and complex data processing.

Secondly, the emergence of unmanned aerial vehicle (UAV) remote-sensing technology provides new opportunities for water quality monitoring. UAVs are highly maneuverable, providing possibilities for further optimization of remote-sensing technology. The technical process of water quality monitoring using UAV remote sensing generally includes the following steps: selecting suitable UAVs, sensors, and remote-sensing software for data collection based on factors such as water type, research purpose, and data accuracy [19]; planning the flight path of the UAV based on the shape and size of the study area; using the spectral sensor to acquire water reflectance spectral data and record spatial coordinates after UAV flight; and importing acquired spectral data into remote-sensing software for data processing and analysis. In this process, the corresponding water quality parameters are obtained by selecting the appropriate inversion model and using prior information or field observation data for correction and validation [20,21]. The obtained water quality parameters through remote-sensing technology can be used to evaluate and analyze the pollution status and dynamic changes of the water body. In addition, it is necessary to visualize and output the processing results in the form of graphs or charts. UAV hyperspectral remote-sensing technology can analyze the reflectance spectra of water surfaces to obtain water quality parameter information [22]. Chlorophyll-a is a chemical substance used to determine the presence of blue-green algae and green plants, and its concentration in the water can be determined by identifying the chlorophyll absorption peaks in the spectrum [23]. Suspended solids refer to tiny particles or microorganisms dissolved in water, and their concentration affects the transparency and chromaticity of the water, which can be determined by analyzing the scattering and absorption peaks in the spectrum [24]. Dissolved organic matter refers to organic substances dissolved in water, and its content can reflect the nutritional status and pollution level of the water, which can be determined by analyzing specific absorption peaks in the spectrum [25]. PH value is a characteristic parameter for measuring the acid-base property of water, and its value can be determined by analyzing the pH-sensitive region in the spectrum [22].

Thirdly, there is an urgent need for fast-processing algorithms. Currently, there are some UAV hyperspectral water quality monitoring algorithms available. The algorithm based on partial least squares regression (PLSR) processes the hyperspectral data into a data matrix and uses the PLSR model to fit the experimental data for rapid and accurate prediction of parameters such as water color, blue-green algae, and turbidity [26,27]. The algorithm based on principal component analysis (PCA) and support vector machine (SVM) first uses the PCA algorithm to reduce the dimensionality of the hyperspectral data and convert it into a 2D image [28,29]. Then, the SVM model is used to classify the image and identify parameters such as blue-green algae, green algae, and yellow-brown algae in the water. The algorithm based on multivariable linear regression (MLR) processes the hyperspectral data into a feature matrix and uses the MLR model to predict parameters such as total suspended solids and chlorophyll-a in the water [28]. These algorithms cannot effectively search, extract, integrate, and organize information during the information extraction process, which is an important factor affecting the efficiency of information extraction. By using various band combination algorithms to find the required information, selecting and extracting accurate information, and integrating multiple pieces of information, the efficiency of information extraction can be improved.

To address the above issues, this article conducted a series of research work, this paper selects four key river sections and establishes a water quality hyperspectral monitoring model suitable for engineering application according to the actual needs of water quality monitoring in Xiong’an New Area in winter [30]. The algorithm design is divided into four stages: analysis, synthesis, summary, and application. (1) The corresponding relationship between 18 kinds of transformation data and the content of spectral data is analyzed, and the best accuracy of band difference and band ratio model is obtained; (2) The relative optimal model of each water quality indicator is obtained by integrating the model precision, and the change law of water pollutants is obtained in the form of spatial distribution mapping; (3) The pollutant content of different river sections is summarized, and the potential pollution source types are obtained; (4) Relevant conclusions can be applied to pollution source monitoring, water self-purification capacity assessment, water quality assessment, and other fields. Relevant achievements can promote the progress of water quality surveys from digital to intelligent and promote the development of digital intelligent environmental protection.

## 2. Materials and Methods

### 2.1. The Study Area

Baiyang Lake is a natural lake in the middle of the Daqing River Basin and one of the few lakes on the North China Plain (115°45′~116°07′ E, 38°44′–38°59′ N) (Figure 1) [31]. The lake has a circumference of 215 km, an east–west length of 39.5 km, south–north width of 28.5 km, a water area of 108.8 km^2^, a water level of 7.09 m, a total area of 336 km^2^, and a water storage capacity of 102.4 million m^3^, which located in the fan edge depression at the intersection of Yongding River and Hutuo River alluvial fans in front of Taihang Mountains. The lake receives nine large rivers from the north, west, and south into the lake, such as Baohe River, Tanghe River, Caohe River, and Chulong River. It flows into the Daqing River through the flood gate and overflow weir in the northeast of the lake and the Zhaowangxin River [1].

Four channel segments, A, B, C, and D, were selected for data acquisition and analysis in order to compare the water quality in the lake area (Figure 2) [32]. Channel segment A is the Xiaobai section in the project of diverting the Yellow River into Hebei to fill the lake, which supplies the Yellow River water to Baiyang Lake through the Xiaobai River. The water from the Yellow River Diversion Gate in Puyang City, Henan Province, is a self-flowing water diversion, which flows through 23 counties in Henan and Hebei provinces and finally enters Baiyang Lake (Figure 3a). Channel segment B is the key monitoring section of Guangdian Zhangzhuang National Control Station and the traffic terminal from the water town to the county seat. There are freshwater aquaculture and natural fishing industry, reed and cattail processing industry, labor export industry, fishing tool production, and other production enterprises around, which are at high risk of pollution [33]. The selection of this river section is of great significance for mastering the current situation of water quality in densely populated areas (Figure 3b). Channel segment C is the key monitoring section of Anzhou Automatic Station and is one of the important tributaries upstream of Baiyang Lake. It passes through a large number of agricultural lands, residential lands, and production lands, and its water quality is directly related to the water quality of Baiyang Lake (Figure 3c). Channel segment D is the key section of Nanliuzhuang National Control Station, where the upstream water flows into Baiyangdian Lake. The river section is divided into paddy fields. Due to the use of chemical fertilizers and pesticides, the potential pollution risk is high (Figure 3d).

### 2.2. Sensor and Data Processing

The GaiaSky-mini2-VN hyperspectral imaging system of Dualix Spectral Imaging Company, which located at 58-1-108 Feihong Road, Nanhu Avenue, Liangxi District, Wuxi City, Jiangsu Province, was used to obtain winter hyperspectral data of four channel segments. The sensor is a high-performance airborne hyperspectral imaging system developed for small rotorcraft. The sensor adopts the built-in scanning system and stability enhancement system with independent intellectual property rights, which overcomes the problems of poor imaging quality caused by the vibration of the UAV system when the small UAV system is equipped with a push-scan hyperspectral camera [34].

GaiaSky-mini2-VN uses the built-in push-scan imaging method. The width can reach 234 m, and the spatial resolution can reach 0.27 m at a flight height of 500 m. The spectral resolution is 2.50 nm, and 176 band data can be obtained between the range of 400~1000 nm. The effective spectral resolution is 4 nm [35]. The M600 pro-UAV system produced by DJ Company is adopted, which integrates high-stability image stabilization platform, data-acquisition controller, and high-precision positioning device, and can realize real-time acquisition and storage of high-quality spectral data, and data processing after returning to the ground [36]. The hyperspectral data of four channel segments in the study area were obtained in four periods. The data is processed according to the process of data restoration, reflectance calculation, and geometric correction in order to obtain the true reflectance of the water body. Data restoration is to restore the original data collected by the hyperspectral camera to uncompressed 16-bit hyperspectral data, image fast-view image and calculated reflectivity data [37]. Reflectivity calculation is mainly based on extracting the reference spectral curve from the diffuse reflectance target cloth laid on the ground during the same flight and then combining it with the known calibration file of the diffuse reflectance target cloth, calculating the reflectance of the spectral data collected during the flight to obtain the hyperspectral reflectance data [38]. Calibration correction mainly uses the original hyperspectral data to carry out dark pixel removal, relative radiation correction, and spectral calibration by using the calibration file. Geometric calibration is conducted to extract the position and attitude data from the inertial navigation system, synchronize the hyperspectral data, use the position and attitude parameters to perform geometric correction on the hyperspectral data, and eliminate the geometric deformation caused by camera tilt and attitude instability [30,39]. Moreover, it synchronizes the hyperspectral data, sets the reference projection plane of the WGS84 coordinate system, uses the position and attitude parameters to geometric correct the hyperspectral data, eliminates the geometric deformation caused by camera tilt and attitude instability, and outputs the reflectivity data with GPS position information under the WGS84 coordinate system according to the position and attitude data output by the inertial navigation system [25,35].

**Figure 3 sensors-23-04089-f003:**
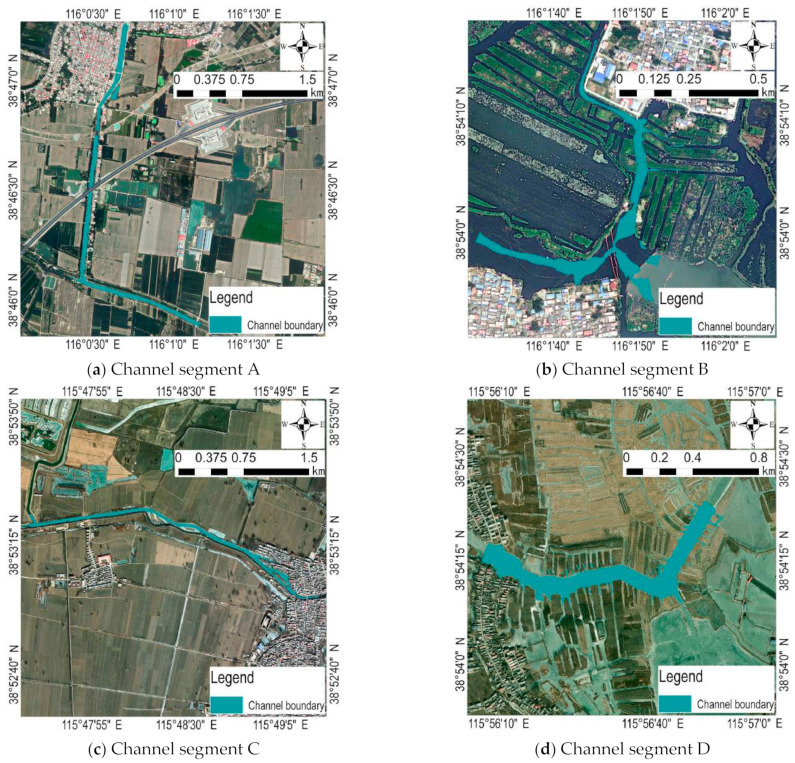
Geographic location and water boundary of the four river sections. (**a**) A water area of 3.48 km in length and 0.21 km^2^ in area was obtained, and the perimeter of the water boundary was 15.08 km on 5 November 2022; (**b**) B water area with a length of 2.05 km and an area of 0.05 km^2^ was obtained, and the perimeter of the water boundary was 4.62 km on 20 December 2022; (**c**) C water area with a length of 3.12 km and an area of 0.13 km^2^ was obtained, and the perimeter of the water boundary was 9.08 km on 17 January 2023; (**d**) D water area with a length of 1.82 km and an area of 0.19 km^2^ was obtained, and the perimeter of the water area boundary was 3.66 km on 24 February 2023.

On the day of hyperspectral data acquisition of UAVs, water samples are collected synchronously (Table 1) [28]. Take water samples from each sampling point with a 400 mL polyethylene bottle, seal, and store them in a box containing an ice bag. The COD, PI, AN, TP, and TN were obtained within 6 h. A total of 184 points of laboratory data were obtained. COD, TP, and TN are measured by the DR6000 spectrophotometer produced by HACH [9]. The assay accuracy of TP can reach 0.01 mg/L, and the measurement accuracy of TN and COD is 0.1 mg/L. The PI is measured by HACH’s COD-203A instrument and the oxidation reduction potentiometric titration method [32]. The principle of ammonia nitrogen measurement is that the ammonia nitrogen in the form of free ammonia or ammonium ion reacts with Nessler’s reagent to form a reddish-brown complex. The absorbance of the complex is proportional to the content of AN [40]. The absorbance is measured at 420 nm [41].

### 2.3. Algorithms

Two core ideas of algorithm design are adopted in order to realize the fast application of water quality hyperspectral algorithm. The first step is to carry out a series of transformation processing on the spectral data, including six forms, exponential, multivariate scattering correction, envelope removal, logarithm, homogenization, differentiation, etc. The second step is to expand the scale to the first and second order on this basis and further explore the relatively optimal inversion model (Table 2) [30,40,42].

After calculation, it is equivalent to 17 kinds of spectral extended data in addition to the original spectral data. In Python program, the mathematical model of content and spectral characteristic value is established. In order to facilitate the realization of the model and meet the application requirements of terabyte big data, two band combination algorithms, band difference and band ratio, are selected to carry out the inversion and accuracy evaluation of key water quality parameters [43]. The band difference model is a classic model for selecting characteristic bands, which can obtain the bands that have the closest relationship with water quality content and the bands that have the least close relationship [31]. By removing interference information, a more accurate calculation model can be obtained. The formula is
(1)$$y=a\left(F_{1}-F_{2}\right)+b,$$
where y is the inverse value of water content; a and b are model coefficients, respectively; $F_{1}$ is the characteristic variable most relevant to the content; and $F_{2}$ is the characteristic variable most irrelevant to the content. In addition to the subtraction model, the accuracy of the ratio model is higher for the low content level where the content is in the percentile or even the thousandth [36]. In the case of poor atmospheric correction effect, the ratio model can further remove the radiation correction error between bands and improve the retrieval accuracy of the model. The formula is
(2)$$y=a\left(\frac{F_{1}}{F_{2}}\right)+b,$$
where y is the inverse value of water content; a and b are model coefficients, respectively; $F_{1}$ is the characteristic variable most relevant to the content; and $F_{2}$ is an arbitrary combination of two spectral characteristic variables.

### 2.4. Accuracy Evaluation

The *R^2^* is the square value of the correlation coefficient, which is used to evaluate the overall predictive ability of the model. The larger the value, the higher the degree of explanation of the independent variable to the dependent variable and the higher the percentage of the change caused by the independent variable in the total change [32]. If the determination coefficient *R^2^* calculated by the model is closer to 1, it means that the accuracy of the model is higher. For example, if the correlation coefficient between the content of COD and the reflectance data at 540 nm is 0.70, then *R^2^* is 0.49, that is, 49% of the content of COD can be determined by the reflectance at 540 nm [44]. The calculation formula is as follows:(3)$$R^{2}=1-\frac{\sum_{i=1}^{n}{\left(y_{i}-\hat{y_{i}}\right)}^{2}}{\sum_{i=1}^{n}{\left(y_{i}-\bar{y}\right)}^{2}},\;$$
where *n* is the sample size, $y_{i}$ is the assay value of the content of point *i,* $\hat{y_{i}}$ is the content prediction value of spectral method of point *i*, and $\bar{y}$ is the mean of the assay value of the samples. The RMSE indicates the stability of the prediction performance of the model. It represents the degree of dispersion of the model prediction results compared with the true value of the dependent variable. The lower the value, the better the stability of the model prediction results [32,45,46]. The calculation formula is as follows:(4)$$RMSE=\sqrt{\frac{1}{n}\sum_{i=1}^{n}{\left(y_{i}-\hat{y_{i}}\right)}^{2}},$$
where *n* is the sample size, $y_{i}$ is the assay value of the content of point *i*, and $\hat{y_{i}}$ is the content prediction value of the spectral method of point *i*. In general, the closer the slope of *R^2^* and the fitting equation is to 1, the smaller the *RMSE* is, the higher the accuracy of the model is, and the more similar the trend of the prediction result is to the real situation. In order to avoid the overfitting of the training model to misjudge the results, this study screened the best model in different models based on *R^2^* and evaluated the stability of the model through *RMSE*.

## 3. Results

### 3.1. Calculation Results

Linear models have many advantages in the calculation of water quality high-spectral analysis compared to complex models [47]. Firstly, they are relatively simple and easy to understand, making them a good starting point for water quality statistical analysis. Secondly, they are computationally efficient and can be analyzed without the need for iterative algorithms, saving computing resources [48]. Thirdly, they can provide interpretable results by estimating the effect size of each spectral data on water quality parameters. Fourthly, they can handle both continuous and categorical predictors, making them a versatile tool for many applications. Finally, they can be combined with other advanced statistical methods, such as regularization and variable selection, to improve performance and generalizability [29]. These advantages make linear models a popular and practical technique in the application of water quality high-spectral assessment. The spectral data and content data are modeled to obtain the spectral transformation, calculation model, and accuracy evaluation results of the four river sections (Table 3). The results show that the spectral transformation methods required for the same water quality index are different [9]. For example, the conversion methods of COD in four river sections are logarithm, envelope determination, original spectrum, and multiple scaling correction after second-order differential.

The relatively optimal model is obtained from the original spectrum to the complex transformation method. There is no uniform transformation method for the other three indicators. It shows that due to the complexity of water quality and the complexity of the environment when data are obtained, there is no universal spectral transformation method that can highlight the information on water quality parameters [9]. The research in this field is still in its infancy, and it is possible to make a new breakthrough with the increase in data volume and the deepening of the summary of spectral laws [47]. In terms of the calculation model, the mode of COD and PI models is ratio type, while the mode of band combination of AN, TP, and TN is difference type, with similar laws. However, the selected wavelengths are not consistent [42]. For example, the characteristic wavelengths of COD in 4 river segments are 590 nm and 695 nm; 552 nm and 720 nm; 532 nm and 726 nm; and 521 nm and 752 nm, respectively. The other four indicators also present a similar situation. Therefore, this method selects the relatively optimal characteristic band from the statistical law. In terms of model accuracy R^2^, the extraction accuracy of COD is between 0.78 and 0.89; the extraction accuracy of PI is between 0.79 and 0.93; the extraction accuracy of AN is between 0.72 and 0.87; the extraction accuracy of TP is between 0.85 and 0.90; and the extraction accuracy of TN is between 0.82 and 0.91. It can be proved that the most difficult indicators to extract are COD and AN [49]. The characteristic band of the former is mainly in the ultraviolet band. It is difficult to extract COD content using this sensor. The content of the latter is mixed with various types of nitrogen elements, and the dispersion of the content value is not large, so it is difficult to extract. TP and TN are extracted in four river sections with good results [5]. The law of RMSE is mainly related to the distribution range of sample point content value. The larger the data value range, the smaller the RMSE value and the more accurate the model. When the accuracy of the quantitative evaluation results is similar, it is advisable to choose a model with a simpler structure in order to achieve higher computational efficiency. Research has shown that the designed ratio model and interpolation model can achieve fast water quality parameter calculations.

### 3.2. Mapping

The spatial distribution map of five water quality parameters of four channel segments is made on the basis of the model.

Spatial distribution of COD (Figure 4a) [50]. The range of content in channel segment A is between 1.00 and 33.75 mg/L, with an average content of 16.55 mg/L. At the east–west and south–north halves of the river course, the content of COD is relatively low, ranging from 11.00 to 17.73 mg/L. The high content appears downstream near Baiyang Lake and reaches the peak at the south–north half, then slowly decreases, and the content at the north side of the sluice further decreases to the low level. The content value range of channel segment B is between 1.58 and 13.00 mg/L, with an average content of 4.77 mg/L. The closer to the bank, the higher the content of COD, between 7.24 and 1.16 mg/L, and the lower the content in the middle of the river. An obviously high-value area appears in the area where the river meets, and the north is close to the powerhouse. It is speculated that it is caused by the adverse emission or diffusion of certain pollutants. The content range of channel segment C is between 0.10 and 22.00 mg/L, with an average content of 8.23 mg/L. The closer to the bank, the higher the content of COD, which is between 9.89 and 18.56 mg/L, and the lower the content in the middle of the river. There is an obviously high-value area near the living area of residents. It is speculated that it is caused by the adverse emission or diffusion of certain pollutants. The content range of channel segment D is between 4.06 and 5.56 mg/L, with an average content of 4.38 mg/L. In the west of the channel, the content of COD is relatively high, ranging from 4.79 to 5.10 mg/L, while in the east of the channel, the content is relatively low. There is an obvious high-value area near the residential and the riverside areas. The content in the south of the river is significantly higher than that in the north.Spatial distribution of PI (Figure 4b) [51]. The value range of channel segment A content is between 1.08 and 8.00 mg/L, with an average content of 4.82 mg/L. The content in the east–west direction of the river is stable between 3.40 and 4.99 mg/L, and the high content appears in the upstream part of the north–south direction, reaching about 6.00 mg/L, and after entering the downstream, the content is stable at 4.00 mg/L. It is worth noting that the PI of the dam annex before entering the lake further increased. The value range of B content in the channel segment is 1.86~14.00 mg/L, with an average content of 4.17 mg/L. The overall distribution in the river channel is relatively uniform and lower than 8.00 mg/L. In the river channel near the village, in the south and the village in the north, there is a certain high-value area, which is directly related to the diffusion of the river flow. The PI is significantly lower in reach with good mobility. The range of C content in the reach is between 1.200 and 12.00 mg/L, with an average content of 3.03 mg/L. The overall distribution in the river channel is relatively uniform and lower than 6.00 mg/L. There are certain high-value areas at the edge of the river channel and near the confluence of tributaries, which are directly related to the diffusion of river flow. The PI is significantly lower in reach with good liquidity. The range of D content in the reach is between 3.70 and 5.19 mg/L, with an average content of 4.15 mg/L. The content in the west section of the river is significantly higher than that in the east section, and the PI is significantly higher as it is closer to the residential area. The content in the south of the river is generally higher than that in the north.Spatial distribution of AN (Figure 4c) [41]. The value range content of channel segment A is between 0.02~0.42 mg/L, with an average content of 0.09 mg/L. The overall content of the river channel is low, and the content of the north–south upstream is slightly higher. In general, the content rate near the bank and at the river bend is higher than that of other river sections. The value range of content in channel segment B is 0.027~0.20 mg/L, with an average content of 0.04 mg/L. The overall content of the river is low, with a slight increase in the narrow tributaries in the north and near the bank in the south. In general, the content near the bank and at the confluence of the river is slightly higher than that of other river sections, which is at a low value. The content range of channel segment C is between 0.001 and 2.28 mg/L, with an average content of 0.12 mg/L. The overall content of the river channel is low. In the west, the content is higher than that in the east, but the overall content is in a lower range. In general, the content near the bank and at the confluence of the river is slightly higher than that of other river sections, which is at a low value. The content range of channel segment D is 0.024~0.17 mg/L, with an average content of 0.079 mg/L. The overall content of the river channel is low. In the west, the content is higher than that in the east, but the overall content is in a lower range. In general, the content near the bank and at the confluence of the river is slightly higher than that of other river sections, which is at a low value as a whole.Spatial distribution of TP (Figure 4d) [52]. The content range of channel segment A is between 0.003 and 0.20 mg/L, with an average content of 0.05 mg/L. The overall content of the river is low, and the content of the east–west and south–north rivers upstream is slightly higher. Before entering the lake, with the self-cleaning of the river, the content decreases to below 0.02 mg/L. The content range of channel segment B is 0.0001~0.05 mg/L, with an average content of 0.02 mg/L. The overall content of the river is low, and the high value appears at the confluence of the river and near the north wharf, as well as at the two branches in the east. The total phosphorus content of other rivers is lower than 0.015 mg/L, which is in a very low pollution concentration range. The content range of channel segment C is between 0.01 and 0.17 mg/L, with an average content of 0.05 mg/L. The overall content of the river channel is low, and the high value appears at the confluence of the river channel and near the bank, as well as at the residential area in the east. The total phosphorus content of other rivers is lower than 0.016 mg/L, which is in a very low pollution concentration range. The content range of channel segment D is 0.017~1.10 mg/L, with an average content of 0.052 mg/L. The overall content of the river channel is low, and the high value appears at the confluence of the river channel and near the bank, as well as at the residential area in the east. The total phosphorus content of the east river is lower than 0.064 mg/L, which is within a very low pollution concentration range.Spatial distribution of TN (Figure 4e) [53]. The content range of channel segment A is 0.04–0.80 mg/L, with an average content of 0.12 mg/L. The overall content of the river is relatively low, and the content of the east–west and north–south rivers upstream is relatively lower. Before entering the lake, the content shows a slight upward trend, and the content rises to about 0.25 mg/L. The content range of channel segment B is between 0.001 and 0.10 mg/L, with an average content of 0.04 mg/L. The overall content of the river is low, and the distribution is very uniform. The north of the river is close to the powerhouse, and there is a certain high value. It is speculated that the flow velocity is slow at this place, resulting in the enrichment of total nitrogen, and the content increases to more than 0.40 mg/L. The content range of channel segment C is between 0.001 and 8.56 mg/L, with an average content of 3.50 mg/L. The overall content of the river is low, and the distribution is very uniform. The north and south sides of the river are close to the bank, and the east side of the river has entered the residential area, showing a certain high value. It is speculated that the flow velocity is slow at this place, resulting in the enrichment of total nitrogen, and the content of this place increases to more than 0.20 mg/L. The content range of channel segment D is 0.52~2.01 mg/L, with an average content of 0.63 mg/L. The overall content of river channels is low, and the content of river channels in the west is generally higher than that in the east. There are certain high values on the north and south sides of the river, near the bank, and at the residential area on the west side, but the overall content is low, within 1.00 mg/L.

## 4. Discussion

### 4.1. The Relationship between Water Flow and Water Quality

The water quality indicators are extracted based on UAV hyperspectral data, which provides technical support for the establishment of an intelligent water quality monitoring platform in the Xiong’an New Area [25,34,54]. The self-purification of rivers mainly includes dilution, sedimentation, microbial decay, and oxygen-consumption reoxygenation [32,44]. On the one hand, the pollutants can be volatilized, neutralized, and degraded due to the comprehensive effects of physical, chemical, and biological actions after the water is polluted [55]. On the other hand, microorganisms in water can decompose pollutants in water to purify water quality [56]. In general, pollutants in water bodies are affected by the geographical environment, hydrological conditions, species and quantity of microorganisms, water temperature, reoxygenation capacity, nature of pollutants, the concentration of pollutants, etc. These processes occur simultaneously and affect each other [9]. Therefore, the traditional dotting sampling method is difficult to achieve, and the introduction of new technology is urgently needed in order to accurately grasp the water quality change of the river [38,57].

The hyperspectral sensor is carried on the UAV, and the data is obtained along the river channel. The assessment data of various pollutants in the river channel can be obtained at one time in a short time [34,35,58]. Taking COD as an example, the calculation results of each channel centerline are extracted from the inversion results (Figure 5). The water quality distribution presents four laws:

Uniform type. Water pollutants are evenly distributed throughout the river, indicating that there are no obvious sewage outlets along the coast, or the pollutants in the whole river are relatively high, and the self-purification effect is not significant, resulting in the accumulation of pollutants (Figure 5a);Enhanced type. It is a common form of water pollutant enrichment, which often occurs in rivers with many pollution discharge points along the coast. With the gradual discharge of pollutants, the self-purification capacity of the river is exceeded, resulting in serious pollution of the river water (Figure 5b);Jitter type. There are sporadic pollution sources along the river. New pollutants will flow in but not exceed the self-purification capacity of the river after a period of self-purification of the river. It shows a fluctuating distribution (Figure 5c);Weakened type. When the pollutants at the source of the river are high, or there are clean tributaries flowing in along the way, the river will show a gradual decrease in the content of pollutants. Under the influence of the self-purification capacity of the river water, the downstream water quality is significantly improved (Figure 5d).

**Figure 5 sensors-23-04089-f005:**
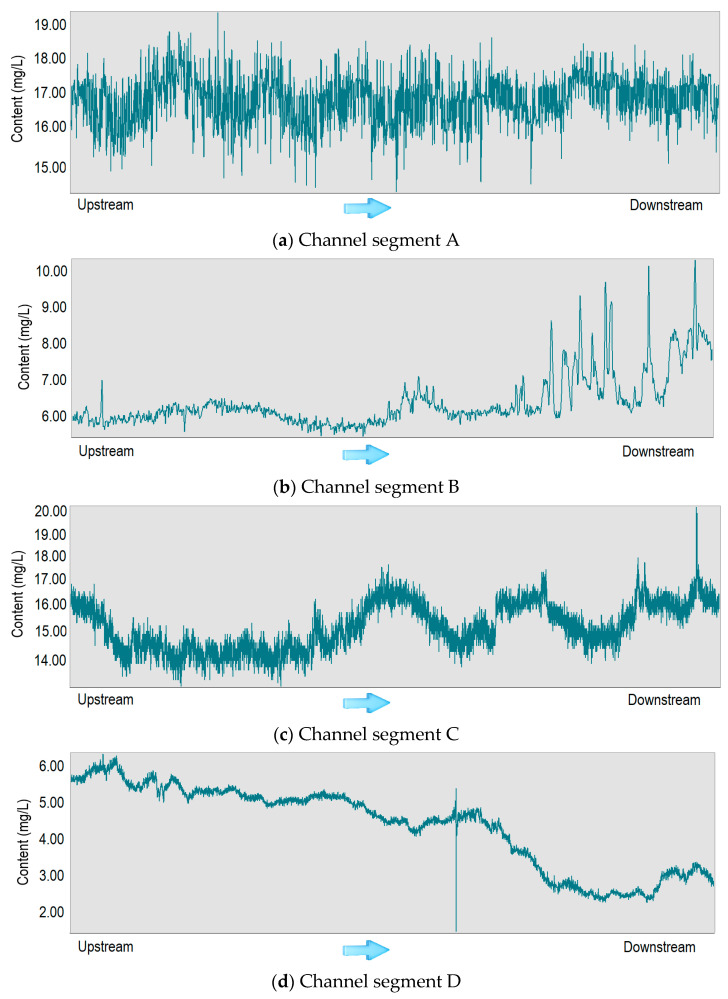
Relationship between COD content and flow direction in different channel segments. (**a**) The pollutant content in segment A has little to do with water flow; (**b**) The pollutant content gradually increases as the water flows downstream; (**c**) The content of pollutants shows an alternating pattern of increasing and decreasing; (**d**) Contrary to channel B, the pollutant content gradually decreases as the water flows downstream.

### 4.2. Frontiers of Hyperspectral Water Quality Algorithms

With the gradual development of the construction of the Xiong’an New Area and the work of draining water into the lake, there is an urgent need for a technology that can efficiently and quickly monitor the water quality and evaluate the health of the water environment [2,3,44]. In terms of instrument principle, water-quality-monitoring technology can be divided into contact technology and non-contact technology. The former includes the water probe method, determination method, and biological method; The latter includes remote-sensing spectroscopy, laser method, and transmission method. Each method has its scope of application and shortcomings [28]. For example, the water inlet probe needs to wipe the sensor regularly, the chemical method will produce secondary pollution, and the biological method has no quantitative ability [56]. Therefore, non-contact optical methods have gradually become the technical trend. Research shows that UAV hyperspectral technology can play a precise application effect in chlorophyll a, suspended particulate matter, dissolved organic matter, transparency, total phosphorus, total nitrogen, ammonia nitrogen, biochemical oxygen demand, water color, colored dissolved organic matter, dissolved organic carbon, transparency, pH, turbidity, and water depth [41,46,57,59,60].

The latest frontier of water quality spectral research includes the use of remote-sensing technology for water quality monitoring and analysis. These cutting-edge technologies will help to deepen our understanding of the causes and evolution of water pollution problems and promote water quality protection and governance. High-resolution, multi-spectral remote-sensing data can be obtained in large water areas using UAV hyperspectral remote-sensing technology, which reveals the presence and distribution of various organic and inorganic substances in water bodies [22,24,27,61]. At the same time, related mathematical models can be constructed to invert water quality parameters and discover new sources of pollution from remote-sensing data for real-time monitoring and warning of the water environment. In addition, chemical mapping analysis based on spectral analysis and chemometrics theory has been widely applied in the field of water quality detection, providing new ideas and methods for water quality evaluation [25,62]. Ultraviolet-spectral-analysis-based water-quality-monitoring technology also has significant advantages such as real-time detection, strong targeting, high accuracy, and low cost [27,63].

In UAV remote-sensing modeling, commonly used modeling algorithms include SVM, random forest, KNN, and ANN [29,45,64,65,66]. Each algorithm has its own advantages and disadvantages, and suitable algorithms need to be selected according to specific problems and data features. This study improves the generalization ability and accuracy of the model by transforming the original spectral data, which can handle nonlinear problems and high-dimensional data, save a lot of computing time and computing resources, and reduce the interference of noisy data [17,29]. Differential and ratio models were designed to classify water quality due to factors such as insufficient sampling samples on the water surface and atmospheric interference, enabling the algorithm to train models quickly and perform classification. UAV hyperspectral water-quality-monitoring technology is a new method for water quality monitoring. Its advantage lies in the ability to efficiently and quickly obtain spectral data on large areas of water bodies, thereby accurately predicting and monitoring the pollution status and dynamic changes of the water. This article conducted research on UAV hyperspectral water quality monitoring in terms of research area design, data acquisition, model building, and data application [26,27]. Compared with existing UAV hyperspectral water-quality-monitoring algorithms based on PLSR, PCA, SVM, and MLR, the computational efficiency has been significantly improved, and the monitoring accuracy for specific water quality parameters has been improved.

Xiong’an New Area is a state-level new area under the jurisdiction of Hebei Province. The importance of water resources is self-evident in maintaining the construction of the future city [2]. Baiyang Lake, the key water body for future urban water use, was selected as the research area, and the water quality of four typical river sections was mainly studied [47,67]. The water quality information extraction was carried out according to five steps, including hyperspectral image data acquisition of UAV, hyperspectral data preprocessing, water spatial information extraction, water quality key index calculation, and water quality mapping and evaluation [54,68,69]. Among them, the extracted indexes include chemical oxygen demand, permanganate index, ammonia nitrogen, total phosphorus, and total nitrogen. This technology breaks through the key technology of UAV hyperspectral water-quality-monitoring, promotes the development of water quality surveys from digital to intelligent, and promotes the progress of digital intelligent environmental protection [38,58]. With the maturity of technology, new technology in the field of water quality surveys will develop in the direction of informatization, objectivity and intelligence [51,61,70]. The research results provide a scientific basis for water source traceability assessment, water pollution source area analysis and water environment comprehensive treatment.

## 5. Conclusions

UAV hyperspectral remote sensing for water quality monitoring is one of the current hot research topics in the field of water quality monitoring. In response to the shortcomings of traditional water-quality-monitoring methods, UAV hyperspectral remote-sensing technology can effectively obtain remote-sensing data on large-area water bodies, improving monitoring efficiency and accuracy. With the continuous development of UAV and hyperspectral technologies, more and more new algorithms have been proposed, such as deep learning models, including CNN and RNN, as well as optimization models based on machine learning and genetic algorithms [17,23]. These new algorithms can comprehensively utilize remote-sensing data from different bands to improve monitoring accuracy and classification accuracy and achieve more comprehensive and multi-angle water quality parameter monitoring and identification. In addition, some cutting-edge issues need to be further explored in the field of UAV hyperspectral remote sensing for water quality monitoring, such as how to establish appropriate mathematical models to retrieve water quality parameters and address the challenges faced by water quality parameter retrieval in different regions, how to use remote-sensing data to discover newly discovered pollutants in water environments, and how to integrate artificial intelligence and remote-sensing technology in practical applications [27,71,72]. Solving these problems will help promote the application and development of UAV hyperspectral remote-sensing technology for water quality monitoring in different fields, such as environmental protection, water resources management, and water ecological protection.

As far as we know, this current research has improved our understanding of the relationship between key components in the process of river water transportation from the perspectives of hyperspectral data acquisition, water quality model establishment, chemical element extraction, etc. This research indicates that the GaiaSky-mini2-VN sensor can quickly calculate the water quality evaluation results of the basin under the support of limited water surface sampling data [25]. The band difference and band ratio model can play a very good role in the corresponding water quality parameters. UAV hyperspectral water quality monitoring has the advantages of areal imaging, real-time monitoring, low cost, and no secondary pollution [55]. The spatial distribution of pollutants can be obtained based on the model [73]. In general, the overall water quality of the river section in the study area is Grade III, with good water quality according to the classification method of surface water quality in the Chinese Environmental Quality Standards for Surface Water (GB3838-2002), and the potential risk pollutants are COD and PI. The impact of land use data on water quality assessment is significant. Land use type and intensity affect the characteristics of land surface runoff and hydrological cycles, thereby affecting water quality and the health of ecosystems. For example, urbanization and agricultural expansion increase the level of pollution in water bodies and also increase the input of non-point source pollutants. Therefore, in the process of water quality assessment, the indicator system and threshold of water quality assessment can be adjusted according to the different types and intensities of land use, to fully consider the impact of land use on water quality. However, further research is needed in terms of winter conditions where land data is covered by snow and ice.

## Figures and Tables

**Figure 1 sensors-23-04089-f001:**
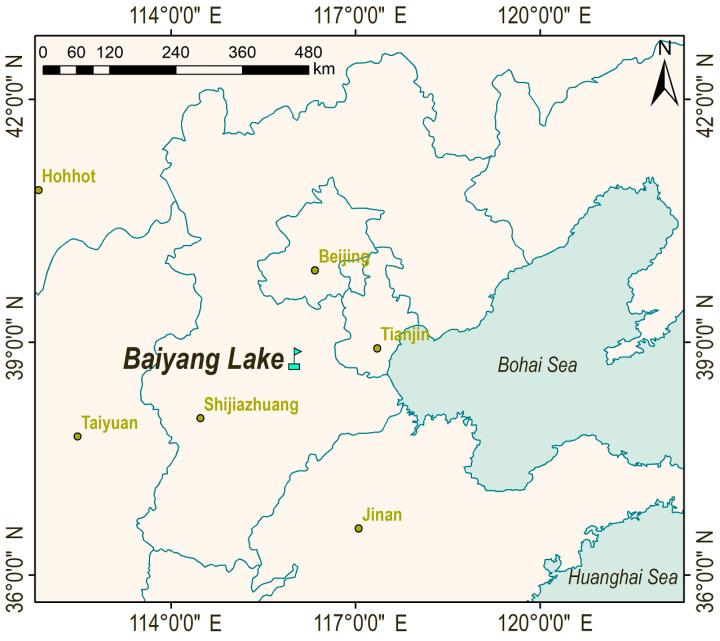
Location of Baiyang Lake. The study area is located 140 km southwest of Beijing, the capital of China. China decided to establish Xiong’an New Area in Xiongxian County, Anxin County, and Rongcheng County on 1 April 2017. Most of Baiyang Lake is under the jurisdiction of Xiong’an New Area and has become an important ecological water body for the development of Xiong’an New Area.

**Figure 2 sensors-23-04089-f002:**
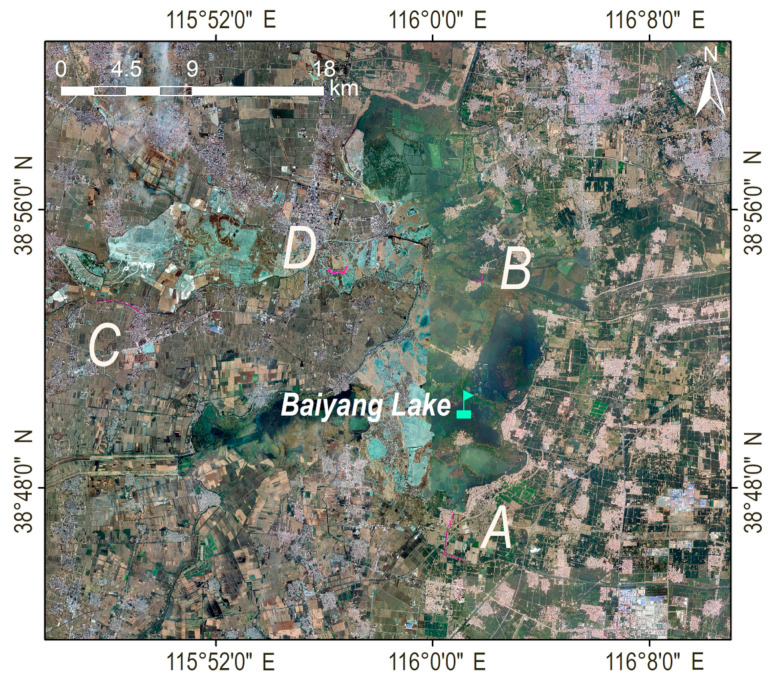
Four channel segments, A, B, C, and D, were selected for data acquisition and analysis.

**Figure 4 sensors-23-04089-f004:**
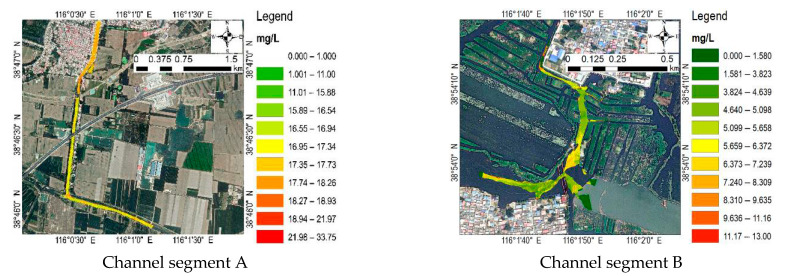
Mapping results of five water quality indicators in four river segments. (**a**) The content of COD indicates the rule of high value in the area where the river meets and is close to the residential area; (**b**) PI is uniformly distributed in the river; (**c**) The AN content near the bank and at the river bend is higher than that in other river sections; (**d**) The overall TP content of the river is low, and the high value occurs at the confluence of the river and near the bank; (**e**) The TN content of the whole river channel is low and evenly distributed.

**Table 1 sensors-23-04089-t001:** Statistical values of water quality parameters of different dates consisting of 184 sampling test data.

Channel Segment	COD (mg/L)		PI (mg/L)		AN (mg/L)		TP (mg/L)		TN (mg/L)	
	Range	Mean	Range	Mean	Range	Mean	Range	Mean	Range	Mean
A	11.20–23.00	17.54	4.20–6.00	4.85	0.03–0.19	0.07	0.07–0.13	0.10	0.04–0.12	0.11
B	2.99–11.89	5.40	1.60–10.21	3.76	0.03–0.12	0.05	0.02–0.05	0.03	0.10–0.67	0.44
C	13.79–20.03	14.34	1.21–11.52	4.03	0.05–2.36	0.12	0.01–0.17	0.05	0.54–8.56	3.51
D	3.94–6.22	4.95	2.65–11.01	4.92	0.12–0.32	0.04	0.02–0.19	0.06	0.61–4.47	1.24

**Table 2 sensors-23-04089-t002:** Spectral transformation methods and calculation formulas. The collected spectral data are processed by a series of spectral transformations to highlight the effective information expression in the spectral data.

Serial Number	Transformation Method	Process Formulas
1	Original spectrum	$X_{i}=R_{i}$
2	Exponential	$X_{i}=e^{R_{i}}$
3	Multiple scattering correction	$X_{i}=\left(R_{i}-b_{i}\right)/k_{i}$
4	Envelope elimination	$X_{i}=R_{i}/C_{i}$
5	Logarithm	$X_{i}=\mathrm{Ln}\;\left(R_{i}\right)$
6	Homogenization	$X_{i}=\left(R_{i}-R_{min}\right)/\left(R_{max}-R_{min}\right)$
7	First-order differential	$X_{i}=R_{i}{}^{’}$
8	Second-order differential	$X_{i}=R_{i}{}^{’’}$
9	Exponential after first-order differential	$X_{i}=e^{R_{i}{}^{’}}$
10	Exponential after second-order differential	$X_{i}=e^{R_{i}{}^{’’}}$
11	Logarithm after first-order differential	$X_{i}=\mathrm{Ln}\;\left(R_{i}{}^{’}\right)$
12	Logarithm after second-order differential	$X_{i}=\mathrm{Ln}\;\left(R_{i}{}^{’’}\right)$
13	Homogenization after first-order differential	$X_{i}=\left(R_{i}{}^{’}-R_{min}^{’}\right)/\left(R_{max}^{’}-R_{min}^{’}\right)$
14	Homogenization after second-order differential	$X_{i}=\left(R_{i}{}^{’’}-R_{min}^{’’}\right)/\left(R_{max}^{’’}-R_{min}^{’’}\right)$
15	Envelope elimination after first-order differential	$X_{i}=R_{i}{}^{’}/C_{i}$
16	Envelope elimination after second-order differential	$X_{i}=R_{i}{}^{’’}/C_{i}$
17	Multiple scattering correction after first-order differential	$X_{i}=\left(R_{i}{}^{’}-b_{i}\right)/k_{i}$
18	Multiple scattering correction after second-order differential	$X_{i}=\left(R_{i}{}^{’’}-b_{i}\right)/k_{i}$

*X_i_* is the processed spectral reflectivity; *R_i_* is the spectral reflectivity; *i* is the band variable; $b_{i}$ is the baseline offset; *k_i_* is the baseline translation; $R_{min}$ is the minimum reflectivity; $R_{max}$ is the maximum reflectivity; $C_{i}$ is the envelope curve value.

**Table 3 sensors-23-04089-t003:** Calculation model and precision evaluation results of water quality key indicators.

Segment	Indicators	Transformation Method	Calculation Model	R^2^	RMSE
A	COD	Logarithm	$y=-459\frac{b_{695}}{b_{590}}+486.90$	0.82	5.39
	PI	First-order differential	$y=-3.20\frac{b_{432}}{b_{790}}+27.00$	0.87	5.63
	AN	Envelope elimination	$y=577\left(b_{484}-b_{552}\right)+11.65$	0.83	0.69
	TP	Exponential	$y=-23.09\left(b_{488}-b_{468}\right)+6.51$	0.85	0.20
	TN	First-order differential	$y=-2.85\left(b_{853}-b_{649}\right)-0.47$	0.82	0.60
B	COD	Envelope elimination	$y=-7.32\frac{b_{720}}{b_{552}}+3.44$	0.89	3.74
	PI	Logarithm after first-order differential	$y=-9.32\frac{b_{769}}{b_{873}}+4.56$	0.92	3.76
	AN	Homogenization after second-order differential	$y=594.31\left(b_{484}-b_{552}\right)-392.40$	0.72	3.78
	TP	Envelope elimination	$y=583.53\left(b_{623}-b_{462}\right)-947.41$	0.90	0.42
	TN	Exponential after first-order differential	$y=-873.40\left(b_{769}-b_{607}\right)-5.32$	0.91	1.24
C	COD	Original spectrum	$y=-9.32\frac{b_{726}}{b_{532}}+5.45$	0.85	1.09
	PI	Multiple-scattering correction after first-order differential	$y=-10.32\frac{b_{765}}{b_{976}}+3.12$	0.79	2.74
	AN	Multiple-scattering correction after first-order differential	$y=398.64\left(b_{945}-b_{594}\right)-8.74$	0.87	0.79
	TP	Multiple-scattering correction	$y=384.13\left(b_{712}-b_{437}\right)-8.69$	0.86	0.73
	TN	First-order differential	$y=-746.30\left(b_{732}-b_{614}\right)-6.62$	0.89	1.49
D	COD	Multiple-scattering correction after second-order differential	$y=2.36\frac{b_{752}}{b_{521}}+5.98$	0.78	0.22
	PI	Logarithm after second-order differential	$y=-2.66\frac{b_{853}}{b_{524}}+1.85$	0.93	3.84
	AN	Multiple-scattering correction after first-order differential	$y=124.63\left(b_{563}-b_{851}\right)-23.32$	0.76	0.31
	TP	Homogenization	$y=263.21\left(b_{462}-b_{836}\right)-52.48$	0.90	3.01
	TN	Homogenization	$y=254.32\left(b_{752}-b_{589}\right)+0.26$	0.84	1.32

*b_i_* is the processed or original spectral data, and *i* is the corresponding wavelength (nm).

## Data Availability

The data and algorithm code presented in this study are available on request from the corresponding author.

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
