# Peer review of "Winter Water Quality Modeling in Xiong’an New Area Supported by Hyperspectral Observation"

_sensors, 2023, doi:10.3390/s23084089_

Round 1

Reviewer 2 Report

The method section needs to be rewritten clarified it is hard to understand.
The result section needs to be reorganised as well.

Reviewer 3 Report

1. The introduction section devotes a lot of space to the background of the article, but there is no mention of the scientific value of the paper and the applicability of the research methods, which are recommended to be added.

2. The introduction lacks the literature review.

3. The section "2.2 Sensor and Data Processing" should focus on the data acquisition process and data information, rather than on the instrumentation.

4. The structure of the conclusion part of the paper is confusing, and the determination of the optimal model based only on the accuracy of a single indicator lacks convincing power. It is recommended to determine the optimal model from both qualitative and quantitative aspects.

5. The discussion section lacks a comparison with the results of other relevant studies and lacks persuasiveness.

6. The conclusion section only highlights the advantages of the research method and does not summarize the results of the study, please improve.
